# Global Sensitivity Analysis of the SCOPE Model in Sentinel-3 Bands: Thermal Domain Focus

**Egor Prikaziuk ***[ID] **and Christiaan van der Tol**[ID]

Faculty of Geo-Information Science and Earth Observation (ITC), University of Twente, P.O. Box 217,
7500 AE Enschede, The Netherlands; c.vandertol@utwente.nl
*   Correspondence: e.prikaziuk@utwente.nl or prikaziuk@gmail.com; Tel.: +31-534-897-112

**Abstract:** Sentinel-3 satellite has provided simultaneous observations in the optical (visible, near infrared (NIR), shortwave infrared (SWIR)) and thermal infrared (TIR) domains since 2016, with a revisit time of 1–2 days. The high temporal resolution and spectral coverage make the data of this mission attractive for vegetation monitoring. This study explores the possibilities of using the Soil Canopy Observation, Photochemistry and Energy fluxes (SCOPE) model together with Sentinel-3 to exploit the two sensors onboard of Sentinel-3 (the ocean and land color instrument (OLCI) and sea and land surface temperature radiometer (SLSTR)) in synergy. Sobol' variance based global sensitivity analysis (GSA) of top of atmosphere (TOA) radiance produced with a coupled SCOPE-6S model was conducted for optical bands of OLCI and SLSTR, while another GSA of SCOPE was conducted for the land surface temperature (LST) product of SLSTR. The results show that in addition to ESA level-2 Sentinel-3 products, SCOPE is able to retrieve leaf area index (LAI), leaf chlorophyll content (Cab), leaf water content (Cw), leaf senescent material (Cs), leaf inclination distribution (LAD). Leaf dry matter content (Cdm) and soil brightness, despite being important, were not confidently retrieved in some cases. GSA of LST in TIR domain showed that plant biochemical parameters—maximum carboxylation rate (Vcmax) and stomata conductance-photosynthesis slope (Ball-Berry m)—can be constrained if prior information on near-surface weather conditions is available. We conclude that the combination of optical and thermal domains facilitates the constraint of the land surface energy balance using SCOPE.

**Keywords:** Sentinel-3; OLCI; SLSTR; SCOPE model; global sensitivity analysis; top of atmosphere; TOA; thermal infrared domain; TIR; land surface temperature; LST; brightness temperature; BT

## 1. Introduction

The retrieval of vegetation parameters from remote sensing data in a multi-dimensional parameter space is a challenging task [1]. It relies on a physical or statistical relationship between the radiometric observation and the vegetation parameter of interest. Parameters can only be retrieved if this relationship is consistent and unique, and the number of retrievable parameters is typically constrained by the number of independent observations. Understanding the model sensitivity to the observations can help select the parameters that can be retrieved, and identify the need for a priori information.

Global sensitivity analysis (GSA) is a useful method to quantify the relative importance of the inputs to the outputs of models. In contrast to local sensitivity analysis, the method accounts for input parameter interactions and possible output non-linearity [2]. For several radiative transfer models describing the interaction between solar radiation and vegetation canopies, GSAs have been carried out with the aim of assessing retrievability of vegetation properties from optical reflectance. At the leaf level, GSAs have been carried out for the PROSPECT model [3] version 4 [4] and versions 4, 5 and D [5]. Sensitivity of PROSPECT to hyperspectral LIDAR measurements was quantified by Sun

and Liang [6], to close-range imaging spectroscopy (PROCOSINE model) by Jay et al. [7], and the sensitivity of a modified version of PROSPECT model including leaf protein and leaf cellulose and lignin quantification has been carried out by Wang et al. [8]. The sensitivity of the dorsiventral leaf model DLM has been analysed by Stuckens et al. [9]. At the vegetation canopy level, leaf reflectance and transmittance models are usually complemented with a canopy radiative transfer model, such as SAIL [10] for closed vegetation or INFORM for forest [11]. Many use PROSPECT for the leaf, as typically indicated with the prefix 'PRO' in the name of the model. Several GSAs have been carried out for the PROINFORM model [12–14]. In a GSA of PROSAIL and PROGEOSAIL in relation to fuel moisture content, Bowyer and Danson [15] showed that leaf area index (LAI) and fraction of vegetation cover dominated reflectance in shortwave infrared (SWIR) region. This dominance of vegetation coverage is reduced if the sensitivity analysis is carried out separately for sparse, intermediate and dense vegetation [16]. At regional and global level it is useful to carry out a GSA for specific operational satellite sensors or platforms. For example, Bacour [17] conducted a sensitivity analysis of four canopy models, PROSAIL, PROKUUSK, PROIAPI and PRONADI, at POLDER bands on Parasol (decommissioned in 2013) [17]. Other GSAs included MODIS on Aqua and Terra (bands 3–7, 13, 15, 16) [18], TM on Landsat 5 (decommissioned in 2013) [19], ETM+ on Landsat 7 (bands 2–5, 7) [20], MSI on Sentinel-2 [21], REIS on RapidEye [22] and WVC on HJ-1 [23]. When dealing with satellite data it is necessary to account for the effect of the atmosphere. Several studies have quantified the atmospheric effects by simulating the propagation of top of canopy (TOC) reflectance to top of atmosphere (TOA) radiance using the models MODTRAN [24,25] for Hyperion on EO-1 (decommissioned in 2017) [26], CHRIS on Proba-1, TM on Landsat 5 and ASTER on Terra [27] and 6S [28] for VEGETATION on SPOT (decomissioned in 2015) [29], MODIS on Aqua and Terra (bands 1–7) [30] and vegetation indices derived from TM and ETM+ on Landsat 5 and 7 respectively [31]. GSA of full range PROSAIL-MODTRAN TOA radiance spectra (400–2500 nm) has recently been reported [32].

Most radiative transfer models exclusively focus on one spectral domain, i.e., either the visible-SWIR, or thermal, or microwave domain, and thus cannot exploit the combined optical and thermal data provided by some platforms, such as MODIS, Landsat and Sentinel-3. In this respect, the Soil Canopy Observation, Photosynthesis and Energy fluxes (SCOPE) model is a useful tool, as it simulates both the thermal and optical radiance signals from the vegetation as a function of vegetation properties and heat fluxes. In addition, it simulates solar induced chlorophyll fluorescence (SIF). In several studies, the sensitivity of SCOPE was analyzed. A GSA of SIF as simulated by SCOPE has been carried out by Verrelst et al. [33]. Gross primary productivity (GPP) simulated with biochemical part of SCOPE was subjected to GSA by Wolanin et al. [34] and latent heat flux and transpiration from energy balance part were analyzed by Jin et al. [35]. None of these studies included the thermal radiance spectra into GSA, although Bayat et al. [36] with a modified version of SCOPE demonstrated that simultaneous usage of optical and thermal bands of Landsat 5 (TM) and 7 (ETM+) leads to more accurate simulations of GPP and evapotranspiration in the course of the growing season. Thus, there is both a need and an opportunity to assess the sensitivity of the thermal and visible bands of Sentinel-3 with SCOPE, and, to the best of our knowledge, this has not been published before.

The objective of the present study is to analyze the usability of Sentinel-3 derived data to constrain the SCOPE model, using a GSA. This is a step towards estimating energy fluxes from SCOPE using satellite data as input. This could eventually support land surface (e.g., ORCHIDEE [37], Noah-MP [38]) and hydrological models (e.g., MIKE SHE [39] or SimSphere [40]) that do not include a radiative transfer description, and that would otherwise rely on more qualitative, index-based remote sensing data products.

The remainder of the paper is organized as follows: materials and methods (Section 2) describe SCOPE model, Sentinel-3 instruments, simulation of Sentinel-3 optical and thermal signal, basics of GSA analysis and the description of the synthetic TOA dataset on which retrieval was performed. The results (Section 3) show full-spectrum and band specific sensitivity indices for optical domain,

the results of retrieval on the synthetic dataset and sensitivity indices in thermal domain (full-spectrum and land surface temperature). In the discussion (Section 4) the explanation of the results along with the comparison with previously published works is given. Conclusions (Section 5) summarize our work.

## 2. Materials and Methods

### 2.1. SCOPE

The Soil Canopy Observation, Photosynthesis and Energy fluxes (SCOPE) model [41] is a homogeneous (1D) radiative transfer model that simulates soil reflectance, leaf reflectance and canopy reflectance factors in optical domain (0.4–2.4 μm), and canopy and soil emitted radiance in thermal domain (2.5–50 μm), as a result of energy balance closure and leaf temperature calculations. The ability to simulate both optical and thermal spectra makes SCOPE a suitable tool for working with sensors operating in both domains, which, among many others, include MODIS on Terra and Aqua, ETM+ on Landsat 7, TIRS on Landsat 8 and SLSTR on Sentinel-3. The thermal radiation measurements facilitate the retrieval of parameters related to the aerodynamic roughness and evaporative cooling [36,42].

SCOPE has been described elsewhere in detail [41,43]; here we only describe the parts that are essential for the present study. The model consists of radiative transfer modules for the leaf (Fluspect [44]) and for the canopy (optical, fluorescence and thermal domain) and models of soil reflectance (BSM [45,46]), energy balance and biochemistry [47–49]. For this study two radiative transfer modules are most relevant: optical radiative transfer module (RTMo) that, strictly speaking, operates with incident irradiance in both optical and thermal domains and thermal radiative transfer module (RTMt) that calculates canopy and soil thermal emitted radiance.

#### 2.1.1. Optical Radiative Transfer Module (RTMo)

At the first step, leaf reflectance, transmittance and fluorescence are calculated with the Fluspect model based on leaf optical parameters listed in Table 1 [44], and these optical properties apply to all leaves in the vegetation canopy. Soil reflectance is simulated with the BSM (Brightness-Shape-Moisture) model that uses 4 parameters (Table 1) [45,46]. Leaf and soil reflectance are integrated to canopy level by RTMo, a numerical version of the SAIL model [10,50]. RTMo calculates the optical properties of vegetation layers based on a leaf angle distribution (prescribed by two parameters, LIDFa and LIDFb, see [51]), leaf area index (LAI) and illumination-observation geometry: solar zenith angle, viewing zenith angle and the azimuthal difference between illumination and view angles. RTMo outputs four top of canopy (TOC) reflectance factors: bidirectional ($r_{so}$), directional-hemispherical ($r_{sd}$), hemispherical-directional ($r_{do}$), bihemispherical ($r_{dd}$) [52]. TOC reflectance ($\rho_{TOC}$) is calculated from reflectance factors and direct ($E_{dir}$) and diffuse ($E_{dif}$) TOC irradiance following Equation (1):

$$\rho_{TOC} = \frac{E_{dir} \cdot r_{so} + E_{dif} \cdot r_{do}}{E_{dir} + E_{dif}} \qquad (1)$$

#### 2.1.2. Thermal Radiative Transfer Module (RTMt)

Thermal emitted radiance is calculated for each layer (60 layers of sunlit and shaded leaves and a layer of sunlit and shaded soil) for wavelength from 2.5 to 50 μm from leaf temperature and emissivity using Planck's law. The leaf temperatures are obtained as a result of energy balance closure, i.e. temperatures are updated iteratively until the net radiation matches with the non-radiative energy dissipation via turbulent heat exchange and evaporative cooling. The emissivity is $1 - \rho - \tau$ for leaf, and $1 - \rho_s$ for soil according to Kirchhoff's law, where $\rho, \tau$ are leaf reflectance and transmittance $\rho_s$—soil reflectance in the thermal range. They are either provided as input to the model or the values at 2400 nm are used for the thermal range as well, because a thermal leaf emissivity model is not

available in SCOPE. Leaf and soil thermal emitted radiance is propagated to canopy diffuse upward and downward radiation and radiance in observation direction in analogy with the optical domain.

### 2.1.3. Energy Balance

The core of the energy balance is the equation:

$$\int_{400}^{50000} (1 - \rho - \tau) \cdot (E_{in} - E_{bb,em}) d\lambda = R_n = H + \lambda E (+G) \tag{2}$$

where $(1 - \rho - \tau)$ - leaf absorptivity (emissivity), $E_{in}$ is the total incident irradiance on a leaf calculated by RTMo and RTMt, $E_{bb,em}$ the emitted black body radiation calculated by RTMt, $\lambda$ the wavelength, $R_n$ is net radiation, $H$ the sensible heat flux, $\lambda E$ the latent heat flux and $G$ the ground (soil) heat flux, which only applies to the soil components but not to leaves.

The sensible and latent heat fluxes are calculated from an aerodynamic resistance scheme. The within-canopy resistance and above canopy resistance are functions of atmospheric stability [53], wind speed, leaf area index and drag coefficients [54]. Finally, stomatal conductance, which is a function of the actual assimilation rate and relative humidity, determines the transpiration rate and latent heat flux. The energy balance routine resolves the temperatures of sunlit and shaded leaves and soil iteratively until Equation (2) holds for all leaves and the soil with a maximum error of 1 W m$^{-2}$. In some cases the algorithm does not converge towards energy balance closure, in which case a warning is issued and the output cannot be accepted. Because the sensitivity analysis scheme that was used in this study does not allow missing values, we gap-filled erroneous values and values where energy balance error (difference between net radiation and the sum of heat fluxes) was higher than 10 W m$^{-2}$ with Gaussian process regression realized in the package `gp_emulator` [55,56].

### 2.2. Sentinel-3

Sentinel-3 is a satellite constellation (currently consisting of two satellites) under ESA's Copernicus program [57]. Both satellites have four instruments on board: Synthetic Aperture Radar Altimeter (SRAL), microwave radiometer (MWR), Ocean and Land Colour Imager (OLCI) and dual-view Sea and Land Surface Temperature Radiometer (SLSTR). The revisit time of the constellation is 1.1 days at the equator. Sentinel-3 has provided level-1 products of top of atmosphere radiance and brightness temperature since 19 April 2016. Although the primary goal of Sentinel-3 is ocean monitoring—topography with SRAL and MWR and temperature with SLSTR—it can well be used for vegetation remote sensing. In 2015 the fluorescence explorer (FLEX) mission was selected to fly in tandem with Sentinel-3 for plant photosynthesis monitoring and fluorescence retrieval [58].

OLCI has 21 bands in the range from 0.4 to 1.02 µm (Figure 1). The swath width is 1270 km with spatial resolution 300 m (full resolution) or 1200 m (reduced resolution) at nadir, while SLSTR is a dual-view spectrometer that has 6 bands delivering radiance (S1–S6) (Figure 1), 3 bands delivering brightness temperature (S7–S9, 3.7, 10.9, 12 µm) and 2 active fire bands (F1,2 3.7, 10.9 µm). The measurements of SLSTR are taken in two views: nadir (swath width 1400 km) and oblique (55°, swath width 740 km). Spatial resolution of S1–S6 bands is 500 m, S6–S9 and F1–2 is 1000 m. In both oblique and nadir view, the measurements are taken in 4 stripes: A: S1–S6, B and C : S4–S6 and I: S7–S9 and F1,2.

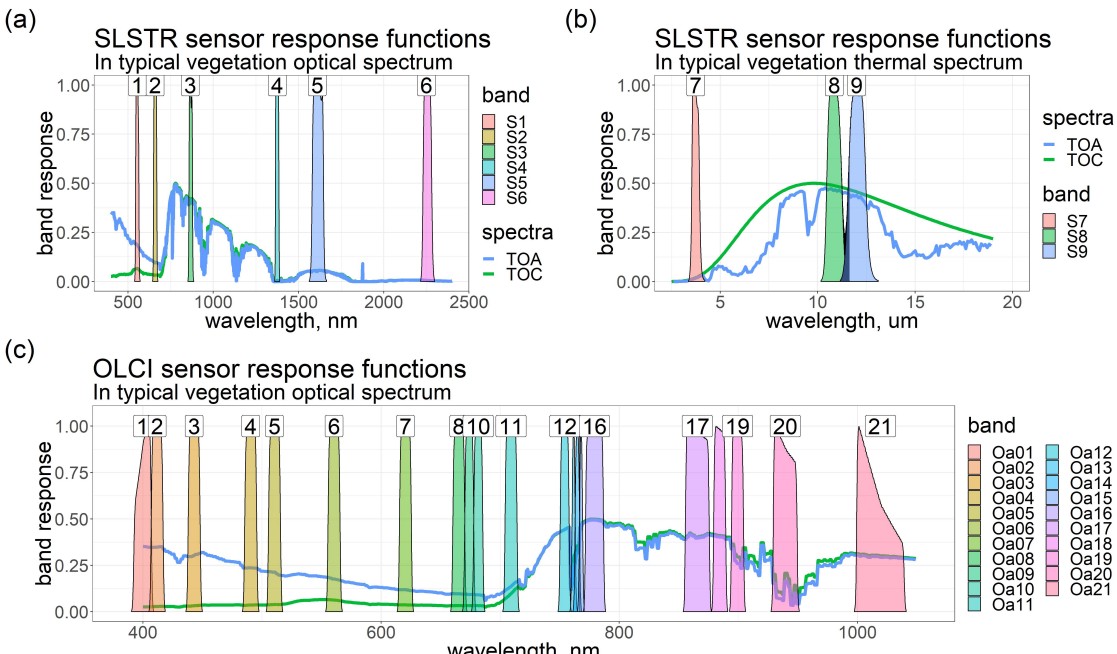

**Figure 1.** Sensor response functions of Sentinel-3 satellite instruments in typical vegetation top of canopy (TOC, green) and top of atmosphere (TOA, blue) radiance spectra. (**a**)—optical Sea and Land Surface Temperature Radiometer (SLSTR); (**b**)—thermal SLSTR; (**c**)—Ocean and Land Colour Imager (OLCI).

### 2.3. Simulation of Sentinel-3 Signal

#### 2.3.1. The Optical Domain (RTMo-6S)

The optical domain includes the 21 bands of OLCI and 12 bands of SLSTR instrument (6 nadir and 6 oblique). Level-1 data expressed in top of atmosphere (TOA) radiance have been available since 19 April 2016, while top of canopy (TOC) synergy data outside the oxygen (Oa13-15) and water (Oa19, Oa20, S4, S4_o) absorption features have been available since 8 October 2018. Because we are interested in the effect of the atmosphere as well, we simulated the Level-1 data. We coupled RTMo module of SCOPE with 6S atmospheric radiative transfer code [28], see Figure 2. TOC reflectance produced by RTMo was transformed into TOA radiance in accordance with the following equation:

$$L_{TOA} = L_{atm} + \frac{E_s \cdot \mu_s \cdot T_g \cdot T_s}{\pi} \cdot \frac{\rho_{TOC}}{1 - S \cdot \rho_{TOC}} \tag{3}$$

where $L_{TOA}$ is the top of atmosphere radiance in observation direction, $L_{atm}$—atmospheric path radiance at TOA, $E_s$—top of atmosphere irradiance for the day of the year, $\mu_s$—cosine of solar zenith angle, $T_g$—total gaseous transmittance, $T_s$—total scattering transmittance, $\rho_{TOC}$—the top of canopy reflectance (Equation (1)), $S$—spherical albedo of the atmosphere. $L_{atm}, T_g, T_s$ are provided in 6S output files, $E_s, \mu_s$ can be calculated (Table A1). For the readers who are familiar with 6S atmospheric correction scheme we provide another version of Equation (3) in terms of 6S atmospheric correction coefficients $xa, xb, xc$ Equation (A1).

Aggregation of 1 nm resolution model output to Sentinel-3 band response was done with spectral response functions of the instruments published by ESA for OLCI (https://sentinel.esa.int/web/sentinel/technical-guides/sentinel-3-olci/olci-instrument/spectral-response-function-data) and SLSTR (https://sentinel.esa.int/web/sentinel/technical-guides/sentinel-3-slstr/instrument/measured-spectral-response-function-data).

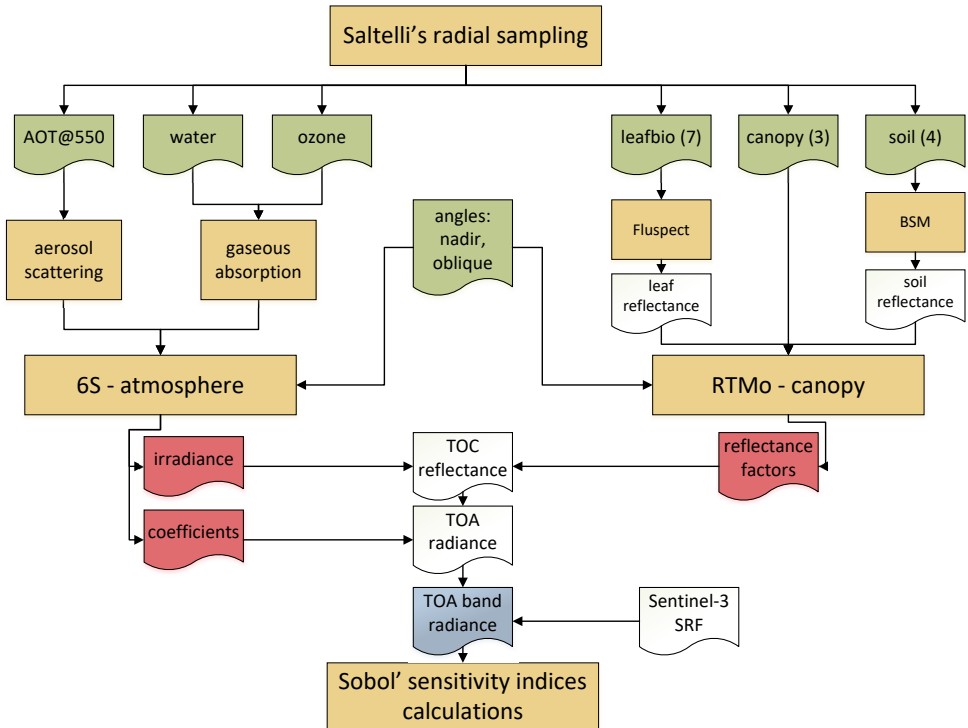

**Figure 2.** Workflow: sampling, model runs, sensitivity indices calculation. Varied input parameters are highlighted in green: atmospheric input consists of 3 parameters: AOT@550, water and ozone; leafbio consists of 7 leaf optical parameters: Cab, Cca, Cdm, Cs, Cw, Cant, N; canopy consists of 3 parameters: LAI, LIDFa, LIDFb; soil consists of 4 BSM model parameters: BSMBrightness, BSMlat, BSMlon and SMC. Notice, that angles (solar and viewing zenith and azimuth angles) were fixed to nadir or oblique configuration. Direct model outputs are highlighted in red. The output used for sensitivity index calculations is highlighted in blue. TOC—Top Of Canopy, TOA—Top Of Atmosphere. Definitions and ranges of input parameters can be found in Table 1.

The Atmosphere (6S)

6S is a radiative transfer model that simulates light propagation through the atmosphere in the range of 0.25–4 μm with the Successive Orders of Scattering method, that is operationally used for the atmospheric correction of MODIS data. In this study, Version 1.1 of S6 was used, wrapped in Py6S package [59]. This wrapper facilitates the input of parameters and provides predefined wavelengths for Sentinel-3 instruments.

Our approach required two groups of 6S output parameters: bottom of atmosphere (BOA) irradiance (direct and diffuse component) for conversion of RTMo reflectance factors to TOC upwelling radiance following Equation (1) and coefficients ($xa$, $xb$, $xc$) for TOC reflectance to TOA radiance propagation (Equation (A1)). We used the 'US62' atmospheric profile, a continental atmospheric model, and a target altitude of 250 m (see Table A2 for complete input). Output parameters and their symbols used throughout this paper are presented in Table A1.

2.3.2. The Thermal Domain (SCOPE)

Thermal products of Sentinel-3 are expressed in TOA brightness temperature (BT) at level-1 and land surface temperature (LST) at level-2. Both products have been available since 19 April 2016. The calculation of LST from BT is carried out with the modified split-window algorithm for bands S8 and S9 following Equation (4) [60,61]. The algorithm includes atmospheric correction:

$$LST = a + b \cdot (BT_{S8} - BT_{S9})^{(1/(\cos(VZA) \cdot m)} + (b + c) \cdot BT_{S9} \tag{4}$$

where $LST$ is the land surface temperature, $BT_{S8}, BT_{S9}$ the brightness temperatures for channels S8 (10.9 µm) S9 (12 µm) of SLSTR; $a, b, c$ are coefficients dependent on plant cover and biome ($a$ also depends on atmospheric water content), $VZA$—satellite viewing zenith angle and $m$—view dependent coefficient.

We simulated top of canopy TOC thermal radiance in observation direction with SCOPE, convolved the thermal radiance to SLSTR S7-9 bands using (the thermal spectral response function, https://sentinel.esa.int/web/sentinel/technical-guides/sentinel-3-slstr/instrument/measured-spectral-response-function-data) and calculated LST by inversion of the Planck equation, acknowledging that the TOC thermal radiance resembles the Planck curve only by approximation, because it is composed of contributions of leaves and soil with different temperature:

$$LST_{S9} = \frac{c_2}{\lambda} \cdot [\ln(\varepsilon \cdot \frac{c_{1L}}{\lambda^5 \cdot L_{S9[TOC]}} + 1)]^{-1} \tag{5}$$

where $LST_{S9}$—equivalent blackbody temperature in $K$, $\varepsilon$—gray body emissivity, $\lambda$—wavelength in m ($12 \cdot 10^{-6}$ m in case of S9), $L_{S9[TOC]}$—top of canopy thermal emitted radiance in observation direction convolved with S9 SRF. The first $c_{1L} = 2 \cdot h \cdot c^2$ and second $c_2 = h \cdot c/k_B$ radiation constants are linear combinations of Planck's constant $h$, Boltzmann's constant $k_B$ and speed of light $c$.

### 2.4. Sobol' Sensitivity Analysis

We employed a Sobol' variance-based global sensitivity analysis as implemented in Python package SALib [62]. This method [63] decomposes total unconditioned variance ($V$) of model output ($Y$), $V(Y)$, into conditional variances that can be attributed to a single parameter $X_i$, such that $V_i = V[f_i(X_i)]$, and parameter interactions of the subsequent orders up to $k$th, where $k$ is the number of model parameters. For example, the variance attributed to the interaction of parameters $X_i$ and $X_j$ is $V_{ij} = V(f_{ij}(X_i, X_j))$. The total unconditioned variance is:

$$V(Y) = \sum_{i=1}^{k} V_i + \sum_{i}^{k} \sum_{j=i+1}^{k} V_{ij} + ... + V_{12...k}. \tag{6}$$

The conditional variance attributed to $X_i$ is the variance of the expected value of $f$ when $X_i$ is fixed and other parameters ($X_{\sim i}$) vary:

$$V_i = V[f_i(X_i)] = V_{X_i}[E_{X_{\sim i}}(Y|X_i)]. \tag{7}$$

The sensitivity index is the portion of total unconditioned variance that can be attributed to the variance caused by the input parameter itself (first-order sensitivity index Equation (8)), caused by its interaction with another input parameter (second-order sensitivity Equation (9)) or caused by the input parameter itself and all its interactions with all other input parameters up to $k$th order (Equation (10)).

$$S_i = \frac{V_i}{V(Y)} = \frac{V_{X_i}[E_{X_{\sim i}}(Y|X_i)]}{V(Y)} \tag{8}$$

$$S_{ij} = \frac{V_{ij}}{V(Y)} = \frac{V_{X_i,X_j}[E_{X_{\sim ij}}(Y|X_i, X_j)] - V_i - V_j}{V(Y)} \tag{9}$$

$$S_{Ti} = S_i + \sum S_{ij} + ... + S_{ij...k} = \frac{E_{X_{\sim i}}[V_{X_i}(Y|X_{\sim i})]}{V(Y)} \tag{10}$$

Efficient calculations of these variances for complex models is commonly done with Monte Carlo integrals [64].

### 2.4.1. Implementation in SALib

The sampling of input parameters was done with `saltelli.sample()` function that constructs quasi-random Sobol' sequence for further calculation of the integrals [65,66] (Appendix B.3), which efficiently explores full parameter space [2]. SCOPE and RTMo were run in Matlab and 6S was run in Python. The sensitivity indices were calculated in Python as well with `sobol.analyse()` (Appendix B.4). Confidence intervals of 0.95 for sensitivity indices were bootstrapped with 100 resampling steps. As our models have many parameters, a threshold of 0.05 for total sensitivity index was taken for classification of parameter impact as significant [67].

### 2.4.2. Parameter Boundaries

We conducted two sensitivity analyses: one for optical and one for thermal domain. In the optical domain the coupled RTMo-6S model was run. The 17 parameters and their ranges are summarized in Table 1. We made 180,000 model runs for OLCI, SLSTR nadir and SLSTR oblique views. In the thermal domain the whole SCOPE model was run. The 32 varied, 4 calculated and 3 fixed parameters are presented in Table 2, whereas the 18 fixed parameters are listed in Table A3. We made 660,000 model runs for SLSTR nadir and SLSTR oblique views. The observation geometry for both models was fixed to simulate nadir and oblique view situation. The angles were not included in the sensitivity analysis because they are always known there is no need to assess their retrievability from observations.

**Table 1.** Ranges of parameters used in global sensitivity analysis (GSA) of coupled RTMo-6S model. Angles in brackets are related to the oblique view. Values for water and ozone are in default 6S units (Sentinel-3 OLCI product units). Fixed 6S parameters are listed in Table A2.

| Parameter | Definition | Unit | Min | Max | Default |
|---|---|---|---|---|---|
| *Leaf (RTMo)* | | | | | |
| Cab | chlorophyll a,b content | $\mu$g cm$^{-2}$ | 0 | 100 | 40 |
| Cca | carotenoid content | $\mu$g cm$^{-2}$ | 0 | 30 | 10 |
| Cant | antocyanin content | $\mu$g cm$^{-2}$ | 0 | 30 | 1 |
| Cdm | dry matter content | g cm$^{-2}$ | 0 | 0.05 | 0.012 |
| Cw | water thickness | cm | 0 | 0.1 | 0.009 |
| Cs | senescent material fraction | - | 0 | 0.9 | 0 |
| N | mesophyll structure | - | 1 | 4 | 1.5 |
| *Canopy (RTMo)* | | | | | |
| LAI | leaf area index | m$^2$ m$^{-2}$ | 0 | 7 | 3 |
| LIDFa | leaf inclination distribution function parameter a | - | -1 | 1 | 0.35 |
| LIDFb | leaf inclination distribution function parameter b | - | -1 | 1 | -0.15 |
| *Soil (RTMo)* | | | | | |
| BSMBrightness | BSM model parameter soil brightness | - | 0 | 0.9 | 0.5 |
| BSMlat | BSM model parameter lat | - | 20 | 40 | 25 |
| BSMlon | BSM model parameter lon | - | 40 | 60 | 45 |
| SMC | volumetric soil moisture content | % | 5 | 55 | 30 |
| *Atmosphere (6S)* | | | | | |
| AOT@550 | aerosol optical thickness at 550 nm | - | 0 | 1 | 0.01 |
| water | atmospheric columnar water pressure | g cm$^{-2}$ (kg m$^{-2}$) | 0 | 7.5 (75) | 1 (10) |
| ozone | atmospheric columnar ozone pressure | atm-cm (kg m$^{-2}$) | 0 | 0.7 (0.015) | 0.326 (0.007) |
| *Angles (RTMo-6S)* | | | | | |
| sza | solar zenith angle | deg | | 50 | |
| oza | observation zenith angle | deg | | 22 (50) | |
| saa | solar azimuth angle | deg | | 150 | |
| oaa | observation azimuth angle | deg | | 100 (195) | |

**Table 2.** Ranges of parameters used in GSA of SCOPE model. Several parameters were calculated. Angles in brackets are related to the oblique view. Fixed SCOPE parameters are listed in Table A3.

| Parameter | Definition | Unit | Min | Max | Default |
|---|---|---|---|---|---|
| *Soil* | | | | | |
| BSMBrightness | BSM model parameter soil brightness | - | 0 | 0.9 | 0.5 |
| BSMlat | BSM model parameter lat | - | 20 | 40 | 25 |
| BSMlon | BSM model parameter lon | - | 40 | 60 | 45 |
| SMC | volumetric soil moisture content | - | 0.1 | 0.7 | 0.25 |
| *Leaf* | | | | | |
| Cab | chlorophyll a,b content | $\mu g\,cm^{-2}$ | 0 | 100 | 40 |
| Cca | carotenoid content | $\mu g\,cm^{-2}$ | 0 | 30 | 10 |
| Cant | antocyanin content | $\mu g\,cm^{-2}$ | 0 | 30 | 1 |
| Cdm | dry matter content | $g\,cm^{-2}$ | 0 | 0.05 | 0.012 |
| Cw | water thickness | cm | 0 | 0.1 | 0.009 |
| Cs | senescent material fraction | - | 0 | 0.9 | 0 |
| N | mesophyll structure | - | 1 | 4 | 1.5 |
| *Canopy* | | | | | |
| LAI | leaf area index | $m^2\,m^{-2}$ | 0.13 | 10 | 3 |
| LIDFa | leaf inclination distribution function parameter a | - | -1 | 1 | 0.35 |
| LIDFb | leaf inclination distribution function parameter b | - | -1 | 1 | -0.15 |
| *Aerodynamics* | | | | | |
| Cd | leaf drag coefficient | - | 0.001 | 1 | 0.3 |
| leafwidth | leaf width | m | 0.001 | 0.5 | 0.1 |
| rwc | within canopy layer resistance | $s\,m^{-1}$ | 0 | 20 | 0 |
| hc | canopy height | m | 0.01 | 50 | 2 |
| rbs | soil boundary layer resistance | $s\,m^{-1}$ | 5 | 30 | 10 |
| rss | soil resistance for evaporation from the pore space | $s\,m^{-1}$ | 100 | 5000 | 500 |
| lambdas | heat conductivity of the soil | $J\,m^{-1}\,K^{-1}$ | 1 | 2 | 1.55 |
| *Biochemical* | | | | | |
| Vcmax (Vcmo) | maximum carboxylation rate at 25 °C | $\mu mol\,m^{-2}\,s^{-1}$ | 0 | 250 | 60 |
| m | slope of leaf conductance-to-photosynthesis | - | 2 | 20 | 8 |
| kV | Vcmax canopy extinction coefficient | - | 0 | 0.8 | 0.64 |
| Rdparam | dark respiration parameter | - | 0.001 | 0.03 | 0.015 |
| *Environment* | | | | | |
| p | air pressure | hPa | 500 | 1030 | 970 |
| rH | relative humidity | - | 0 | 1 | 0.64 |
| u | wind speed at height z | $m\,s^{-1}$ | 0.5 | 10 | 2 |
| Ca | atmospheric $CO_2$ concentration | ppm | 200 | 500 | 380 |
| Ta | air temperature | °C | 5 | 35 | 20 |
| Rin | broadband incoming shortwave radiation (0.4–2.5 um) | $W\,m^{-2}$ | 0 | 1000 | 600 |
| Rli | broadband incoming longwave radiation (2.5–50 um) | $W\,m^{-2}$ | 200 | 500 | 300 |
| *Calculated* | | | | | |
| z | measurement height of meteorological data | m | | $2.5 \cdot hc$ | |
| zo | roughness length for momentum of the canopy | m | | $zo\_and\_d(LAI, hc)$ | |
| d | displacement height | m | | $zo\_and\_d(LAI, hc)$ | |
| ea | atmospheric vapour pressure | hPa | | $rH \cdot satvap(Ta)$ | |
| *Angles* | | | | | |
| tts | solar zenith angle | deg | | 50 | |
| tto | observations zenith angle | deg | | 22 (50) | |
| psi | relative azimuth angle | deg | | 130 (135) | |

## 2.5. Retrieval on Synthetic Dataset

In addition to the GSA, we carried out a retrieval exercise using a synthetic dataset. This dataset consisted of 33 TOA radiance spectra shown in Figure 3.

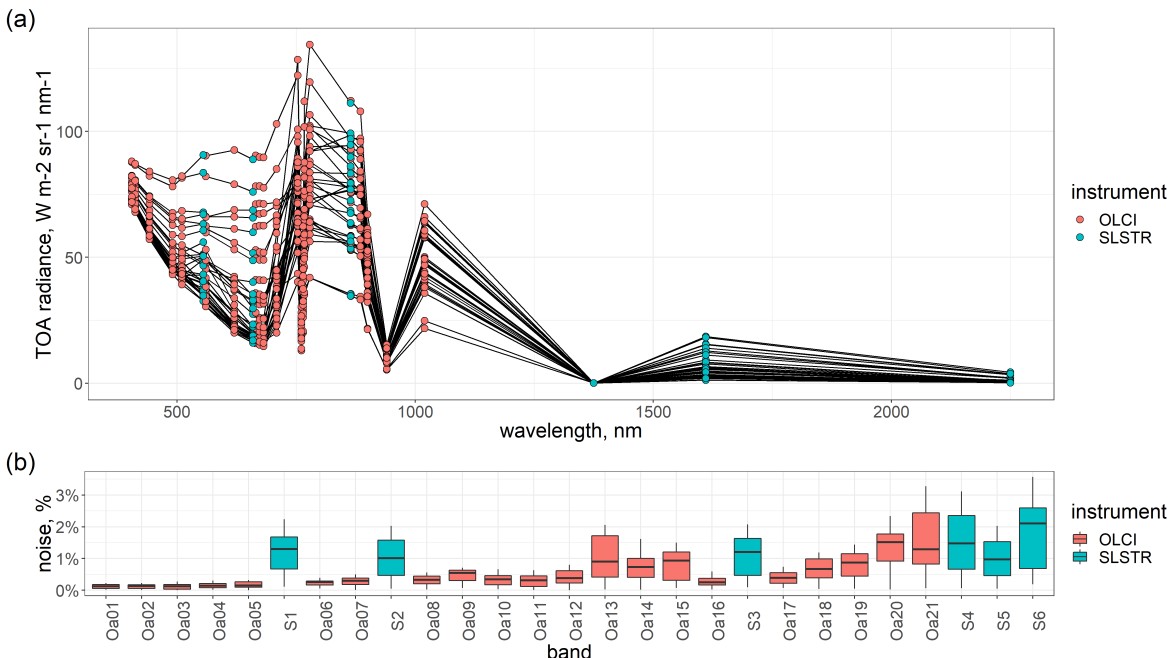

**Figure 3.** (**a**)—33 simulated vegetation TOA reflected radiance spectra in bands of optical Sentinel-3 instruments (OLCI and SLSTR), (**b**)—added noise based on signal-to-noise ratio of the instruments.

TOC reflectance factors were generated with RTMo with all parameters randomly sampled from a uniform distribution within the borders specified in Table 1. TOA radiance was calculated from the resulting reflectance factors with a clear sky, mid-latitude continental atmosphere as Equation (12) in [45]. Finally, random noise ($noise_i$) with average factor of 2.5 was added to the resulting TOA spectra ($signal_i$) based on the signal-to-noise ratio $SNR_i$ of the corresponding band $i$ of OLCI and SLSTR, found in (the Sentinel-3 optical annual performance report of 2018, https://sentinel.esa.int/web/sentinel/user-guides/sentinel-3-olci/document-library/-/asset_publisher/hkf7sg9Ny1d5/content/sentinel-3-optical-annual-performance-report-year-2):

$$noise_i = \frac{signal_i}{SNR_i} \cdot \text{rand}(0,5) \tag{11}$$

The retrieval algorithm is a modified version of the algorithms used in [45,68] that includes BSM model for soil reflectance simulation and is publicly available https://github.com/Prikaziuk/retrieval_rtmo. The retrieval is carried out using the build-in function *lsqnonlin* in the MATLAB Optimization Toolbox that uses the trust-region-reflective method of cost-function minimization, taken into account lower and upper parameter borders [69,70]. The cost function used is a simple difference between modeled and measured spectra. In contrast to [45,68] we did not use prior information on parameter distribution.

## 3. Results

### 3.1. RTMo-6S GSA 400–2400 nm

In this section we present first order sensitivity indices (S1) as a direct effect and total order sensitivity indices (ST) as a combined effect, i.e., appearing from interactions. Following Zhang et al. [67], significant parameters are those for which sensitivity index value is higher than 0.05.

Full-spectra of Sobol' variance-based total order sensitivity indices in the region 400–2400 nm for top of canopy (TOC) reflectance are presented in Figure 4a and for top of atmosphere (TOA) radiance in Figure 4b. The following subsections focus on the results in OLCI and SLSTR bands.

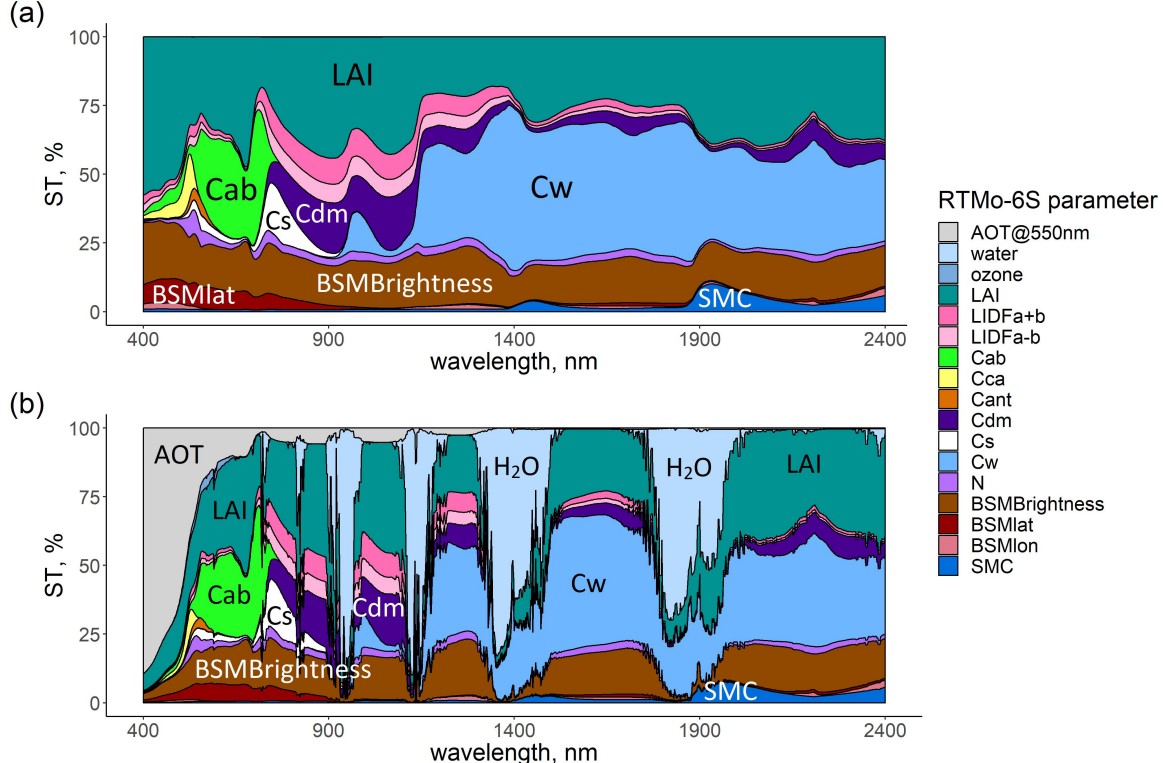

**Figure 4.** Total order sensitivity indices (ST) calculated with coupled RTMo-6S model in the range 400–2400 nm. (**a**)—top of canopy (TOC), (**b**)—top of atmosphere (TOA). ST are expressed in percentages. Definitions of parameters can be found in Table 1.

### 3.1.1. OLCI

Figure 5 shows that a limited number of model parameters significantly influence reflectance in the OLCI bands. The important parameters based on ST value are presented in Figure 5.

All bands in the visible spectral domain (Oa1–10, Figure 5a) are affected by aerosol scattering, especially those in the blue region (Oa1–5). Among primarily important vegetation parameters LAI influences all bands and chlorophyll concentration (Cab) bands Oa6–10. Through interactions carotenoid content (Cca) reaches significance in band Oa5, senescent material fraction (Cs) and leaf mesophyll structure parameter (N) in band Oa6. Parameters of BSM soil model also became important through interaction in bands Oa4–10.

In the NIR spectral domain (Figure 5b) we can differentiate several band groups. First, red-edge band Oa11 has the highest sensitivity to Cab among all OLCI bands, and this sensitivity is even higher than the sensitivity of Oa11 to LAI. Second, the bands located in the oxygen absorption region (Oa12–16) are sensitive to leaf senescent material (Cs), leaf dry matter content (Cdm), leaf angle distribution (LAD driven by LIDFa, LIDFb), LAI and BSMBrightness. Third, bands located further in the NIR (Oa17–19, Oa21) do not show dependence on Cs, but they are still sensitive to Cdm, LAI and LAD. Forth, bands Oa19–20 are located in atmospheric water absorption features and strongly depend on the atmospheric water vapour content parameter of the 6S model. Finally, band Oa21 in the SWIR is sensitive to leaf water content (Cw).

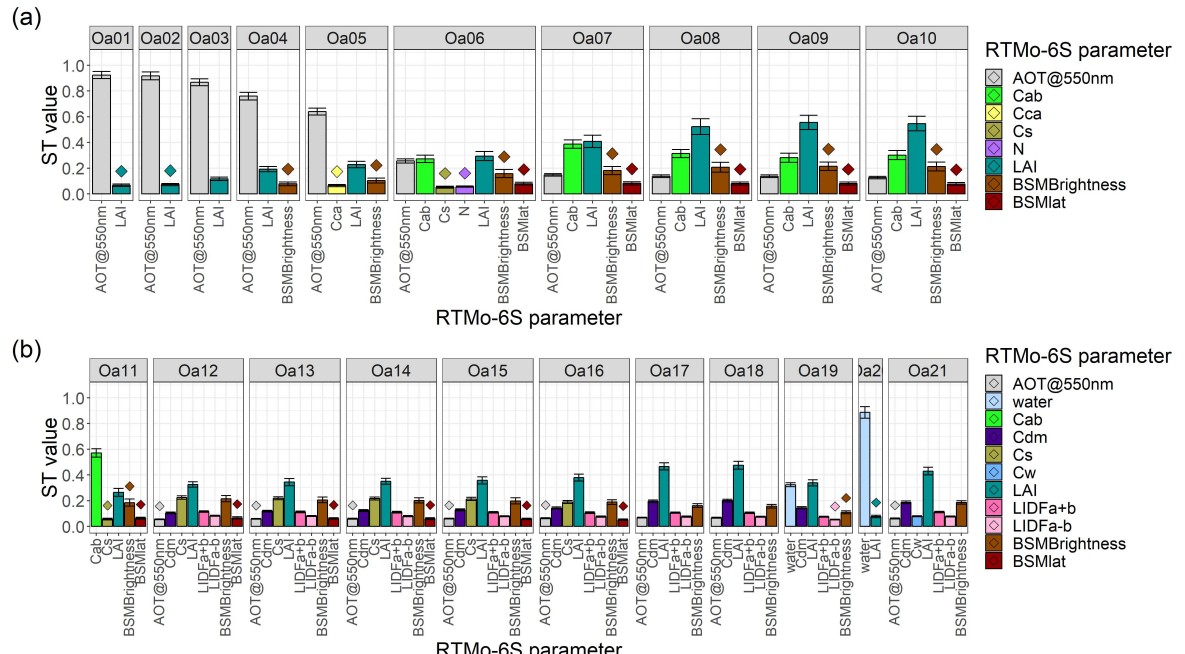

**Figure 5.** Significant total order sensitivity indices (ST > 0.05) for OLCI instrument. Parameters marked with diamonds had insignificant first order sensitivity index (S1 < 0.05) and reached significant ST through interactions. (**a**)—visible domain, (**b**)—near infrared domain. Definitions of parameters can be found in Table 1.

### 3.1.2. SLSTR

The results of GSA in bands of Sentinel-3 SLSTR instrument are shown in Figure 6 for the parameters with ST > 0.05.

The total order sensitivity indices for nadir view of SLSTR instrument (Figure 6a) showed similarities with OLCI bands. The bands in visible range (S1–2) were affected directly by aerosol optical thickness at 550 nm (AOT@550), leaf chlorophyll content (Cab) and LAI and after interactions by BSMBrightness and BSMlat. The band in NIR (S3) in addition to AOT@550, LAI and BSMBrightness showed the effects of leaf dry matter (Cdm) and leaf inclination (LIDFa+b, LIDFa−b). S4 band is located in the atmospheric water absorption feature and hardly contains surface-related information. SWIR bands S5 and S6 had significant influence of leaf water content (Cw).

The panel (b) of Figure 6 shows results for the oblique view of SLSTR. Overall sensitivity index values were similar for oblique and nadir views with the exception of bands S1–3. Band S1 (S1_o) was much more sensitive to AOT@550 and atmospheric ozone concentration in the the oblique view than in the nadir view. Band S2_o also showed higher sensitivity index for AOT@550, whereas band S3_o lost its sensitivity to leaf inclination distribution function parameter a-b (LIDFa-b).

### *3.2. TOA Retrieval (Synthetic Data)*

To prove that GSA results indicate retrievable parameters we conducted the retrieval from synthetic data. The results are presented in Figure 7 and the corresponding metrics are in Table 3. We conducted two sets of retrieval: from OLCI data only (red) and from OLCI and SLSTR data (Synergy, blue). In both cases, the algorithm was able to retrieve most parameters to which the model was sensitive (ST > 0.05), and even some of the parameters with limited sensitivity (ST < 0.05)—Cca, while it was unable to retrieve the input values of LIDFb, Cant, Cdm, N, and all soil parameters (B, BSMlat, BSMlon, SMC). The retrieval from both instruments (Synergy) improved the quality (Table 3).

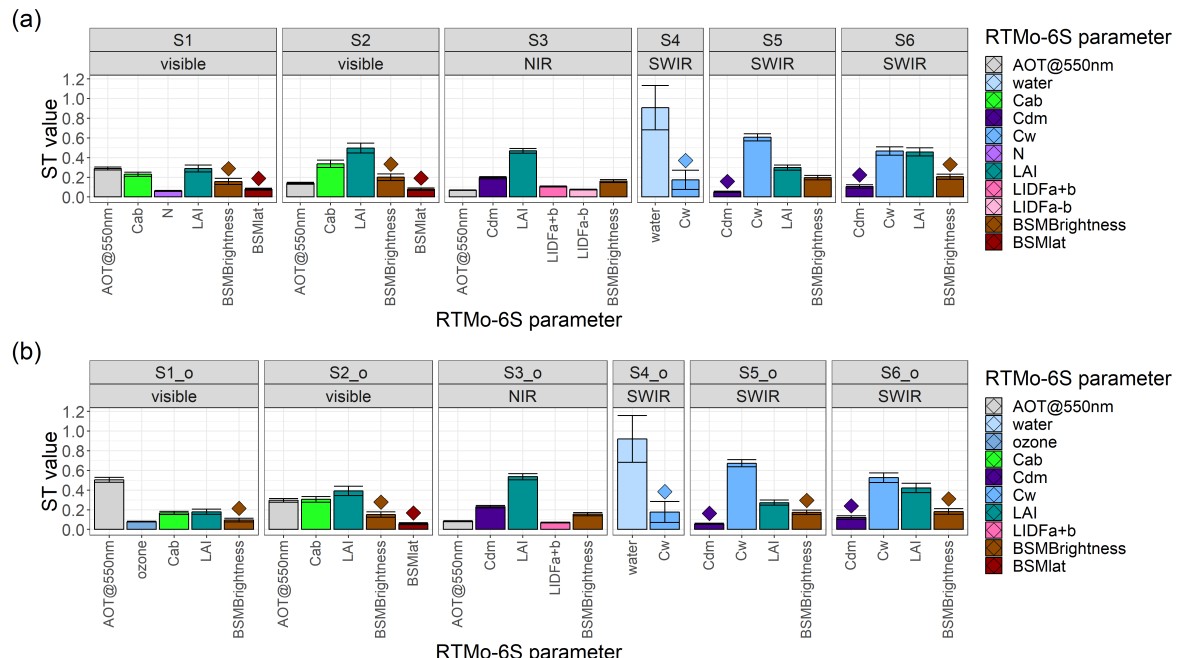

**Figure 6.** Significant total order sensitivity indices (ST > 0.05) for SLSTR instrument. Parameters marked with diamonds had insignificant first order sensitivity index (S1 < 0.05) and reached significant ST through interactions. (**a**)—nadir view, (**b**)—oblique view. Definitions of parameters can be found in Table 1.

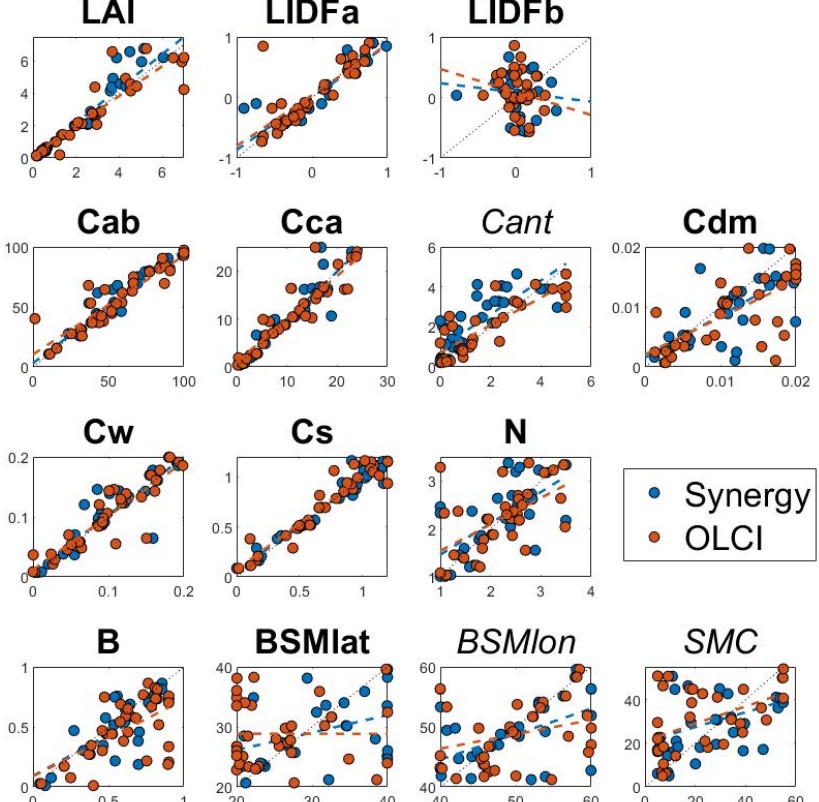

**Figure 7.** Quality of retrieval on a synthetic dataset. Parameters with names in italic (Cant, BSMlon, SMC)—parameters with low sensitivity index values. Top—canopy parameters, middle—leaf parameters, bottom—soil parameters. Red - retrieval from OLCI bands, blue - retrieval from both OLCI and SLSTR (Synergy) bands. Definitions of parameters can be found in Table 1.

**Table 3.** Metrics of retrieval on synthetic dataset. Definitions of parameters can be found in Table 1.

| Metric | Instrument | B | BSMlat | BSMlon | SMC | Cab | Cca | Cant | Cdm | Cw | Cs | N | LAI | LIDFa | LIDFb |
|--------|-----------|------|--------|--------|------|------|------|------|-------|-------|------|------|------|-------|-------|
| RMSE | OLCI | 0.3 | 8.8 | 7.1 | 18.1 | 10.9 | 2.8 | 0.9 | 0.006 | 0.024 | 0.1 | 0.7 | 0.9 | 0.3 | 0.4 |
|      | Synergy | 0.2 | 7.0 | 6.0 | 17.6 | 6.7 | 2.8 | 1.1 | 0.005 | 0.025 | 0.1 | 0.6 | 0.9 | 0.2 | 0.4 |
| RRMSE | OLCI | 53 | 31 | 15 | 61 | 20 | 25 | 43 | 57 | 25 | 15 | 30 | 34 | 742 | 508 |
|      | Synergy | 45 | 24 | 12 | 59 | 12 | 25 | 54 | 48 | 25 | 11 | 27 | 34 | 533 | 532 |
| $R^2$ | OLCI | 0.33 | 0.00 | 0.09 | 0.22 | 0.83 | 0.86 | 0.73 | 0.46 | 0.83 | 0.91 | 0.33 | 0.85 | 0.64 | 0.04 |
|      | Synergy | 0.45 | 0.16 | 0.31 | 0.16 | 0.92 | 0.85 | 0.62 | 0.55 | 0.81 | 0.95 | 0.44 | 0.86 | 0.82 | 0.01 |

### 3.3. SCOPE GSA 2.5–50 um

GSA of the full SCOPE model, i.e., with energy balance, are presented in Figure 8. The first and total order sensitivity indices of top of canopy outgoing thermal radiance in observation direction are presented as percentages. The most influential parameters were wind speed (u), Ball-Berry stomatal conductance parameter (m), maximum carboxylation capacity (Vcmax) and LAI. LAI showed spectral dependence with lower sensitivity below 20 μm and higher towards 50 μm.

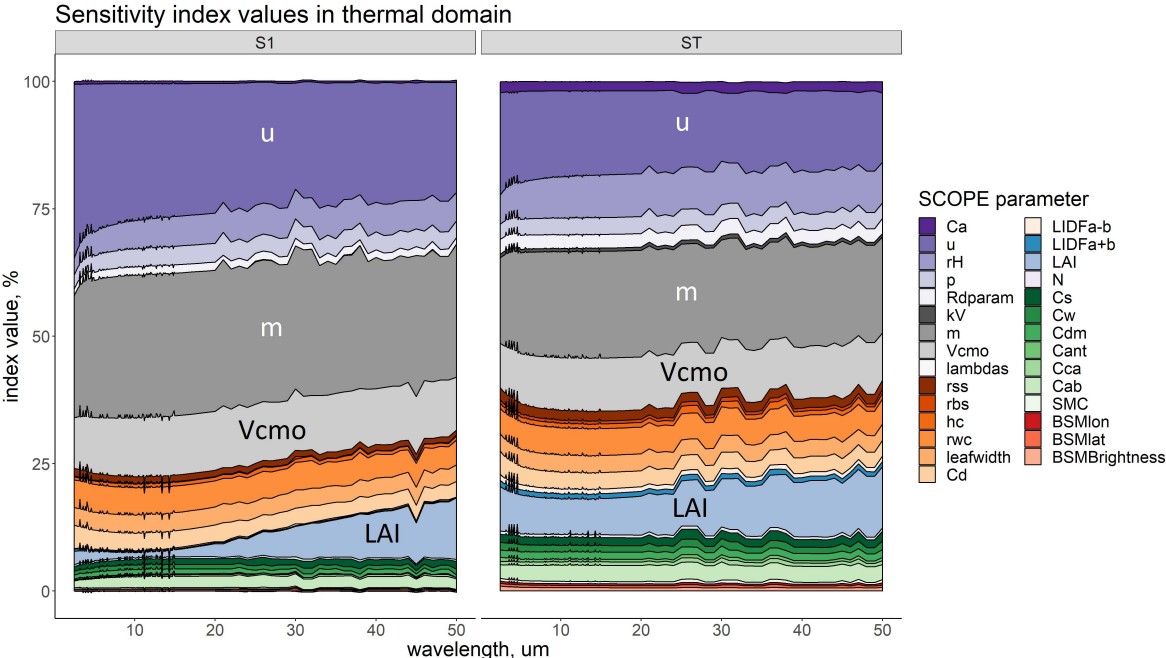

**Figure 8.** First (S1) and total (ST) order sensitivity indices for top of canopy outgoing thermal radiance in observation direction simulated with complete SCOPE model presented as percentages. Definitions of parameters can be found in Table 2.

LST

Figure 9 shows the results of the GSA analysis of the SCOPE model with two sets of parameters: in first instance all inputs were varied (Figure 9a), while in the second instance the near-surface air temperature (Ta) and TOC total incoming short- and longwave radiation (Rin and Rli, respectively) were fixed (Figure 9b). In the first case, the sensitivity of LST is strongly dominated by Ta and TOC irradiance. Indeed irradiance and air temperature are the strongest drivers of surface temperature in the SCOPE model. For an effective use of SLSTR for land surface studies, these will have to be constrained by using data from other sources such as real-time weather data. For this purpose it makes sense to consider the GSA with these weather inputs constant. We fixed their values to Ta = 20 °C, Rin = 600 W m$^{-2}$, and Rli = 300 W m$^{-2}$ (Figure 9b) and repeated the GSA. With this constraint, the biochemical traits—Ball-Berry stomatal conductance parameter (m) and maximum carboxylation capacity (Vcmax)—appeared to influence the SLSTR bands significantly. Among environmental factors wind speed (u), which is related to the aerodynamic resistence to turbulent heat exchange fluxes, showed a high first order sensitivity index value.

After all possible interactions several other model parameters crossed the threshold of significance (ST > 0.05): leaf—leaf chlorophyll content (Cab); canopy—LAI; aerodynamics—leaf drag coefficient (Cd), leafwidth, within canopy aerodynamic resistance (rwc); environment—air pressure (p), relative humidity (rH used for atmospheric vapor pressure (ea) calculations).

Interestingly, only the Ball-Berry stomatal parameter *m* was significantly influenced by the viewing angle Figure A1. Indeed differences in stomatal aperture within the canopy is responsible for a directional differences in brighness temperature in SCOPE.

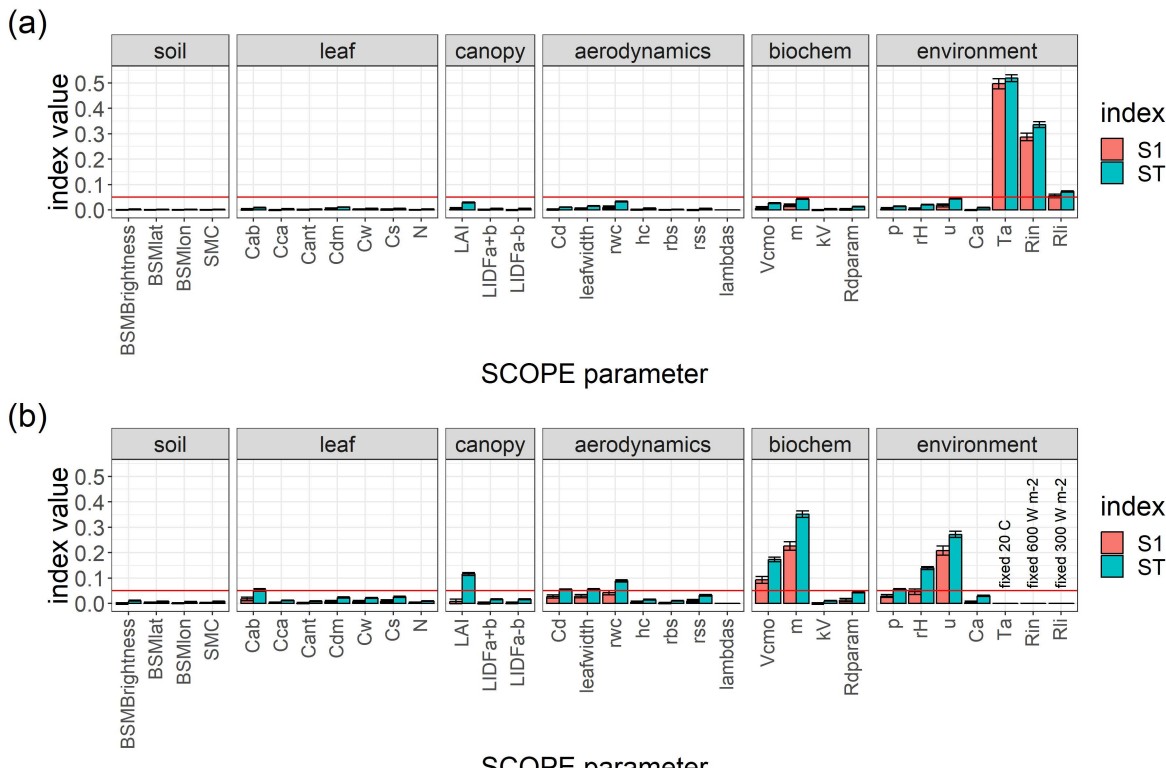

**Figure 9.** First (S1) and total (ST) order sensitivity indices for land surface temperature (LST) simulated with SCOPE model from Sentinel-3 SLSTR instrument bands S8 (10.9 μm) S9 (12 μm). (**a**)—varying all parameters, (**b**)—fixed air temperature (Ta = 20°), total incoming short- and longwave radiation (Rin = 600 W m$^{-2}$, Rli = 300 W m$^{-2}$). Red horizontal line denotes the significance threshold (0.05 units of index value). Definitions of parameters can be found in Table 2.

## 4. Discussion

### 4.1. RTMo-6S GSA 400–2400 nm

Earlier studies have investigated the sensitivity of top of canopy (TOC) reflectance simulated with PROSAIL [15,16,71–73] and the optical domain of SCOPE [33,35] to the model parameters. In general our results agree with the those studies: LAI was a dominant factor throughout the spectral range (400–2400 nm), leaf inclination distribution (LAD) showed slight importance in all wavelength with a substantial peak in 700–1300 nm, leaf chlorophyll content (Cab) affected visible range with peaks in green and red-edge regions, leaf dry matter (Cdm) was important in NIR and SWIR region, leaf water content (Cw)—SWIR region, especially above 1300 nm. Our results were rather different from those of Jin et al. [35] in some other aspects. They found that the sensitivities of SCOPE simulated reflectance to Cab, Cca, Cw and Cdm stretch far beyond the absorption spectra of these components, a finding which we could neither reproduce nor explain theoretically.

Compared to Verrelst et al. [72] who carried out a sensitivity analysis of TOC reflectance simulated by SCOPE, we used revised optical coefficients for pigments in leaves as proposed by Féret et al. [74]

for the PROSPECT-D model, a soil reflectance model below and an atmospheric model above the vegetation. In particular the inclusion of an atmospheric model has a great influence on the model sensitivity [26].

Recent work that included leaf carotenoid content (Cca) into GSA showed a contribution in 400–500 nm region [16,71], and we found that the TOC reflectance output is sensitive to both Cca and the newly introduced anthocyanin content (Cant). However, the sensitivity diminishes when top of atmosphere (TOA) radiance are used, due to the dominance of atmospheric properties in the spectral domain of the pigment absorption. This suggests that only strong constraints on atmospheric parameters will make retrieval of these pigments possible. Because we assumed that the atmosphere was perfectly characterized in the retrieval from synthetic data, the retrieval of these pigments was successful.

Leaf senescent material fraction (Cs) was included in one earlier GSA study [33], and our results are slightly different. Although the spectral range of Cs influence is the same (400–900 nm) our results demonstrated a peak in NIR region whereas, according to Verrelst et al. [33], the highest sensitivity to Cs appeared in the visible range. This difference is caused by the inclusion of a soil model (BSM) in the present study, which lowers the relative contribution of brown leaf material to the total variance.

Our results also showed discrepancies with some other studies in relation to LAI sensitivity. We obtained constantly high sensitivity to LAI all over the optical region, suppressed by aerosol optical thickness at 550nm (AOT@550) in the visible range and interrupted by the water absorption bands. The results of Bowyer and Danson [15] showed declining sensitivity of SAILH to LAI in NIR (800–1000 nm) and according to Yanfang Xiao et al. [16] 4SAIL is not sensitive to LAI changes in the 400–800 nm region at all. The difference between our results and those of these earlier studies is due to the specific combination of models used here, including the soil and atmospheric model, and possibly to the range of LAI parameter space; Yanfang Xiao et al. [16] showed that the influence of LAI on reflectance depends on the vegetation development stage: the influence is higher for sparse canopies (LAI < 3) than for dense canopies.

Recent global sensitivity study of top of atmosphere (TOA) radiance simulated by PROSAIL and MODTRAN showed similar results to our findings [32]. The biggest difference is in the importance of AOT@550—more than 75% in the blue region according to our data and around 30% (AOT@550 and asymmetry parameter G together) in Verrelst et al. [32]. This may be due to the narrower range of AOT@550 used in that study (0.05 to 0.5) compared to our study (0 to 1) and the addition of other atmospheric parameters to GSA—Ångström exponent, single scattering albedo and Henyey-Greenstein (asymmetry) parameter of the phase function. Lower influence of columnar water vapour in our results can be explained by the difference between MODTRAN and 6S algorithms.

### 4.2. OLCI and SLSTR Bands

Our results indicate that LAI and leaf chlorlophyll content (Cab) can be retrieved from visible bands Oa6–10. Red-edge band Oa11 is by far the best choice for Cab retrieval. The Level-2 OLCI terrestrial chlorophyll index (OTCI, https://earth.esa.int/web/sentinel/technical-guides/sentinel-3-olci/level-2/olci-terrestrial-chlorophyll-index) is indeed based on Oa11 in combination with Oa12 and Oa18. NIR bands Oa12–16 are influenced by Cs, leaf dry matter (Cdm), leaf angle distribution (LAD) and soil brightness (BSMBrightness), while Cdm is mostly affected by bands Oa17, 18 and Oa21. As bands Oa1–4 are influenced almost exclusively by AOT@550, they may be used in AOT@550 retrieval algorithms. Oa3 blue band is used in combination with Oa10 and Oa17 for FAPAR estimation—OLCI global vegetation index (OGVI, https://earth.esa.int/web/sentinel/technical-guides/sentinel-3-olci/level-2/olci-global-vegetation-index).

There have been several studies conducting GSA in bands of certain sensors at TOC level: PROSAIL was used for Landsat TM [19], Sentinel-2 MSI [21], REIS of RapidEye [22] and WVC of HJ-1 [23], PARAS model based on the spectral invariants theory was used for Landsat ETM+ [20] and leaf canopy model for several bands of MODIS [18]. At TOA level another soil-leaf-canopy model

with Hapke soil model and two types of leaves (green and brown) was used for Hyperion [26], CHRIS and Landsat TM sensors [27] and a coupled leaf-canopy-atmosphere model where most of the leaf parameters are calculated as percentages of specific leaf weight for MODIS [30]. Due to the narrow bandwidth of Sentinel-3 instruments (average bandwidth is 10 nm) and different models that were used for MODIS instrument direct comparison of the mentioned results with ours is complicated, thus we will discuss only the works devoted to MSI and CHRIS, which have comparable characteristics to Sentinel-3.

The study of MSI was aimed at vegetation water indices assessment with PROSAIL model and GSA was also conducted [21]. Sentinel-2 was designed for vegetation monitoring, it has revisit time of 3–5 days and spatial resolution of 20–60 m. However, its bands are twice as wide as Sentinel-3 and, although band 2 covers carotenoid-dependent region, it is too wide to exhibit enough sensitivity for carotenoind content retrieval. Nonetheless, synergy between Sentinel-2 and Sentinel-3 was discussed in recent studies in relation to OTCI validation with Sentinel-2 [75] and the enhancement of the spatial resolution of SLSTR thermal images (thermal sharpening) for evapotranspiration analysis [76].

CHRIS sensor, the closest in band configuration to OLCI, showed the dominance of LAI over the whole spectral region (400–1000 nm), LIDFa after 700 nm. Two peaks of Cab sensitivity in green (555 nm) and red-edge (700 nm) region for both green and brown leaves and two peaks of Cs at 550 and 730 nm for green leaves only [27]. All these results, besides the peak of Cs in green region, agree with our findings. PROBA-1 satellite with CHRIS on board has a repeat cycle of 7 days.

The signal from SLSTR instrument appears suitable for the retrieval of leaf water thickness (Cw) (S5,6). The optical bands of of SLSTR can further contribute to the retrieval of LAI, Cab, and Cdm, while the off-nadir viewing angle appears most useful promising for the retrieval of AOT@550: In bands S1 and S2 the values of sensitivity indices for oblique view were twice the values for nadir. This agrees with the proposed methodology of AOT@550 retrieval from Multi-angle Imaging SpectroRadiometer (MISR) instrument: larger viewing angles result in longer atmospheric path, thus the signal contains more information about an aerosol haze [77,78]. Level-2 SLSTR products containing AOT@550 will be available in the near future.

### 4.3. TOA Retrieval (Synthetic Data)

The exercise with TOA retrieval showed that most of the parameters with high sensitivity index can actually be retrieved, but the results must be considered with caution due to the fact that synthetic data were used, produced with the model that was also used for retrieval. Moreover, AOT@550 was not retrieved on this exercise.

As an interesting alternative to assessing the sensitivity of a forward model, the sensitivity of an inverted model can be quantified as well. le Maire et al. [79] applied GSA to the inverted PROSAIL model, where MODIS reflectance in red and near-infrared bands was input and LAI was output.

Apart from model sensitivity, parameters also need to have a unique (i.e., distinguishable from that of other parameters) effect on the spectrum in order to be retrievable. This explains why Cca, Cant, and Cw were retrievable from OLCI bands despite limited sensitivity, while the soil parameters BSMBrightness and BSMlat were not successfully retrieved despite a higher sensitivity. From the two leaf orientation parameters, only LIDFa (a measure for the mean leaf angle) was retrievable, while the retrieval of LIDFb (the bi-modality of the leaf angle distribution) was clearly ill-posed, confirming earlier findings by Verhoef et al. [45]. We emphasize that when retrievals are carried out with real TOA measurements rather than synthetic data, model representation errors will affect the results, and their effect may significantly reduce the retrievability below the ideal case shown here.

### 4.4. SCOPE GSA 2.5–50 um

To our knowledge the current work is the first work conducting GSA of a radiative transfer model in the thermal infrared (TIR) domain. Although the simulated emitted thermal radiance spectra is a smooth curve due to the use of spectrally constant emissivity values for soil and leaves, the sensitivity indices showed some spikes. In the region below 15 μm these spikes can be explained by the reflected incoming longwave radiation from the atmosphere, which was the default spectrum in SCOPE as simulated with MODTRAN. This spectrum is not smooth. Spikes at longer wavelengths may be explained by coarse output resolution. The simulated outgoing thermal radiation strictly does not resemble a Planck curve, due to the fact that the outgoing radiation is a linear combination of contributions of individual leaves and the soil with different temperature (which may each separately follow Planck's law). This phenomenon may explain the increasing contribution of LAI along the spectra: leaves are typically cooler than soil and therefore, at higher wavelengths the contribution by leaves to the radiance is higher than at shorter wavelengths.

Components (leaf and soil) temperature and land surface temperature (LST) are used in energy balance models and algorithms for evapotranspiration estimation. Studies that conducted a GSA for LST include the hydrological model MIKE SHE [80], global land surface models ORCHIDEE [81] and Noah-MP [82] and soil-vegetation-atmosphere transfer model SimSphere [83]. Compared to SCOPE these models have a limited description of radiative transfer processes. A comparison between our results and that of previous studies with these models is nevertheless possible.

For SimSphere model outgoing thermal radiation was shown to be dependent on the terrain aspect, LAI and soil moisture content (SMC) [83]. SCOPE does not take the surface slope into account. However, the importance of LAI on thermal radiation was shown in our work as well. SMC did not show significant influence in SCOPE model, because it is used only in optical domain: It affects only shortwave net radiation. SCOPE is not a full SVAT model, and due to the absence of a soil water budget in SCOPE, SMC it does not affect the latent heat flux.

Noah-MP model showed the dependence of soil temperature on canopy height (hc), wind speed (u), LAI and several soil properties [82]. SCOPE model also showed the connection between LST and u and LAI, but not hc. Interestingly, Noah-MP model simulates biochemical processes with a Farquhar-Ball-Berry model that is similar to SCOPE, but in their GSA Vcmax showed relatively little sensitivity, due to dominance of soil related parameters. This can be explained by the explicit connection between soil properties and transpiration in Noah-MP, which is absent in SCOPE.

### 4.5. Limitations

We note that several limitations of the SCOPE model affect the results. The first limitation is the representation of a horizontally layered vegetation in SCOPE. This representation is not accurate for all vegetation types, especially considering the 300–500 m pixel size of the Sentinel-3 data. In heterogeneous land cover, the fractional vegetation cover, which is not included in SCOPE, may be the parameter of great significance [16]. Strategies will have to be developed to unmix vegetation and soil spectra [84].

A limitation of SCOPE in thermal domain is the lack of an emissivity model, and a limitation related to the retrieval is the need for temperature-emissivity separation (TES). Exploiting the different viewing angles [85,86] for either TES or the differentiation between sunlit and shaded soil and leaf temperatures [42] are interesting ideas. However, our GSA showed only very small differences in the sensitivity of (directional) LST between the nadir and the oblique view of SLSTR. Due the swath, the difference in viewing angle is generally less than $50°$, the observations are not close to the hotspot, and for vegetation with higher LAI, the contrast between component temperatures is generally small. These all contribute to the small difference in sensitivities between the two observation directions. Here we considered only TOC radiances, but when the thermal radiance is propagated through the atmosphere, atmospheric effects may dominate the directionality of the observations.

Our results confirm that the information provided by the two instruments on Sentinel-3 is complementary, and suggest that retrievals from the optical domain can help constrain retrievals from the thermal domain. Some parameters affect both the optical and the thermal domain, in particular Cab and LAI. Both influence the absorption of solar radiation, which is a main driver of the surface energy balance, which in turn determines the thermal signature of the vegetation. The two parameters are retrievable from the optical domain, and this retrieval can help constrain retrievals of biochemical parameters such as Vcmax and m from the thermal domain. However, the found connection has to be validated directly with Vcmax and m measurements or indirectly through energy and carbon flux.

## 5. Conclusions

We investigated the possibilities of application of SCOPE model together with Sentinel-3 derived data for vegetation monitoring. In addition to ESA-distributed level-2 products of OLCI global vegetation index (OGVI) for fraction of absorbed photosynthetically active radiation (FAPAR) and OLCI terrestrial chlorophyll index (OTCI), SCOPE provides opportunities for LAI retrieval, leaf dry matter (Cdm) and leaf water content (Cw) retrieval that can be used for fuel moisture content calculations, along with leaf angle distribution parameter a (LIDFa), leaf senescent material fraction (Cs) and soil brightness retrieval.

For the first time, a global sensitivity analysis of the thermal domain of SCOPE model was conducted in application to SLSTR-derived land surface temperature (LST). It was shown that LST had a link to plant biochemical parameters: maximum carboxylation rate (Vcmax) and slope of conductance-to-photosynthesis relationship (m). Limiting uncertainties in environmental factor with prior information from meteorological stations and in plant structural traits from optical information, the retrieval of Vcmax and m parameters is possible.

Our results confirm that the synergistic application of optical and thermal data from Sentinel-3 satellite within a model that coupled both this domains with energy balance and photosynthesis, SCOPE, can be beneficial for the accurate canopy state monitoring.

**Author Contributions:** Conceptualization, E.P. and C.v.d.T.; methodology, E.P. and C.v.d.T.; software, E.P. and C.v.d.T.; validation, E.P. and C.v.d.T.; formal analysis, E.P.; investigation, E.P. and C.v.d.T.; resources, E.P. and C.v.d.T.; data curation, E.P.; writing–original draft preparation, E.P. and C.v.d.T.; writing–review and editing, E.P. and C.v.d.T.; visualization, E.P.; supervision, C.v.d.T.; project administration, C.v.d.T.; funding acquisition, C.v.d.T.

**Funding:** The project has received funding from the European Union's Horizon 2020 research and innovation programme under the Marie Sklodowska-Curie grant agreement No 721995.

**Acknowledgments:** The authors thank Max Planck institute for biogeochemistry for computational support.

**Conflicts of Interest:** The authors declare no conflict of interest.

## Software

| | |
|---|---|
| SCOPE v.1.73 | https://github.com/Christiaanvandertol/SCOPE |
| | docs: https://scope-model.rtfd.io |
| RTMo retrieval algorithm | https://github.com/Prikaziuk/retrieval_rtmo |
| | docs: https://scope-model.rtfd.io/en/latest/retrieval.html |
| Py6S | https://github.com/robintw/Py6S |
| | docs: https://py6s.readthedocs.io |
| SALib | https://github.com/SALib/SALib |
| | docs: https://salib.rtfd.io |
| gp_emulator | https://github.com/jgomezdans/gp_emulator |
| | docs: https://gp-emulator.rtfd.io |

## Appendix A. Results

*LST Difference in Viewing Angles*

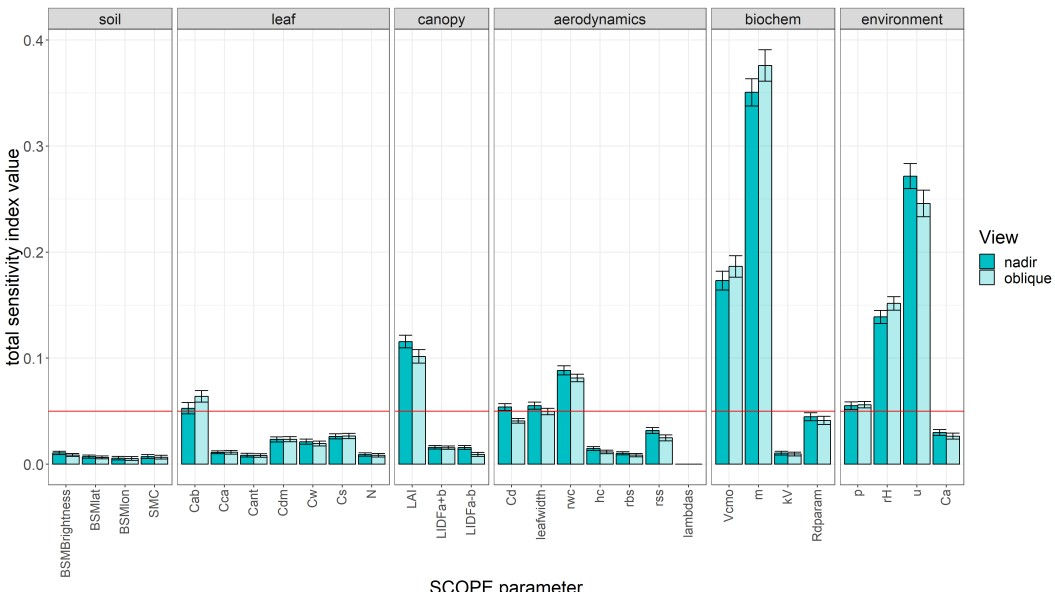

**Figure A1.** Total order sensitivity index (ST) as percent for land surface temperature (LST) simulated with SCOPE model from Sentinel-3 SLSTR instrument bands S8 (10.9 μm) S9 (12 μm) in different views: oblique (transparent), nadir (dense). Red horizontal line denotes the significance threshold (0.05 units of index value). Definitions of parameters can be found in Table 2.

## Appendix B. Materials and Methods

*Appendix B.1. Atmospheric Correction in 6S Atmospheric Model*

In terms of 6S atmospheric correction coefficients the relationship between top of canopy reflectance ($\rho_{TOC}$) and top of atmosphere radiance ($L_{TOA}$) (Equation (3)) can be expressed as:

$$L_{TOA} = \frac{xb}{xa} + \frac{1}{xa} \cdot \frac{\rho_{TOC}}{1 - xc \cdot \rho_{TOC}} \tag{A1}$$

from which one can infer the following definition of 6S coefficients:

$$xa = \frac{\pi}{E_s \cdot \mu_s} \cdot \frac{1}{T_g \cdot T_s} \tag{A2}$$

$$xb = L_{atm} \cdot xa \tag{A3}$$

$$xc = S \tag{A4}$$

where $E_s$—TOA irradiance for the day of the year, $\mu_s$—cosine of solar zenith angle, $T_g$—total gaseous transmittance, $T_s$—total scattering transmittance, $L_{atm}$—atmospheric path radiance at TOA, $S$—spherical albedo of the atmosphere.

*Appendix B.2. Output Parameters of 6S Atmospheric Model*

**Table A1.** 6S atmospheric model output parameters and their symbols used in this paper.

| Symbol | 6S Output Name | Parameter |
|:------:|:--------------:|:---------:|
| $E_{dir}$ | direct solar irr. | bottom of atmosphere direct solar irradiance |
| $E_{dif}$ | atm. diffuse irr. | bottom of atmosphere diffuse solar irradiance |
| $L_{atm}$ | atm. intrin. rad. | atmospheric path radiance (at top of atmosphere) |
| $E_s$ | $\dfrac{\text{int. sol. spect (in w/m}^2)}{\text{int. funct filter (in mic)}}$ | top of atmosphere solar radiance |
| $T_g$ | global gas. trans. total | total gaseous transmittance |
| $T_s$ | total sca. trans. total | total scattering (aerosol) transmittance |
| $S$ | spherical albedo total | spherical albedo of the atmosphere |

*Appendix B.3. Sobol' Parameter Sampling in SALib*

`saltelli.sample()` from SALib Python package creates two equal matrices (A—sample and B—re-sample) of size $N \times k$, where $N$ is the number of simulations and $k$—the number of input parameters. In this first step the matrices are filled with parameters varying between 0 and 1 using bitwise operations. At the second step it resamples matrix A from matrix B with radial sampling [87], resulting in matrix AB ($N \cdot k \times k$). In case of second-order sensitivity index calculations (this work) resampling of matrix B from matrix A was also done, resulting in matrix BA ($N \cdot k \times k$). Overall the sampling matrix is a row-bound matrices A, AB, BA, B of size $N(2k + 2) \times k$. Finally, the matrix of samples is scaled to custom boundaries, given (in our case) uniform distribution.

*Appendix B.4. Sobol' Integration in SALib*

`sobol.analyse()` from SALib Python package computes first-order, second-order and total sensitivity indices taking into account the quasi-random order of input parameters. Output matrix ($N(2k + 2) \times 1$) is normalized and rearranged into matrices A ($N \times 1$), AB ($N \times k$), BA ($N \times k$), B ($N \times 1$). $i$th column of the matrix AB ($AB_i$) has output where values of $X_i$ are equal to values of $X_i$ in matrix A, whereas the rest of the parameters ($X_{\sim i}$) vary. With help of these matrices variances and expectations from Equations (8) to (10) are calculated in SALib in accordance to Equations (A5) to (A7).

$$V_{X_i}[E_{X_{\sim i}}(Y|X_i)] = \text{mean}(B \cdot (AB_i - A)) \tag{A5}$$

$$V_{X_i, X_j}[E_{X_{\sim ij}}(Y|X_i, X_j)] = \text{mean}(BA_i \cdot AB_j - A \cdot B) \tag{A6}$$

$$E_{X_{\sim i}}[V_{X_i}(Y|X_{\sim i})] = \frac{\text{mean}(A - AB_i)^2}{2} \tag{A7}$$

## Appendix C. Fixed Model Parameters

*Appendix C.1. 6S Fixed Parameters*

**Table A2.** Input parameters of 6S atmospheric model that were fixed during the global sensitivity analysis. Angles in brackets are related to the oblique view. Varied 6S parameters are listed in Table 1.

| Parameter | Definition | Unit | Value |
|---|---|---|---|
| *Geometrical conditions* | | | |
| sza | solar zenith angle | deg | 50 |
| saa | solar azimuth angle | deg | 150 |
| oza | observation zenith angle | deg | 22 (50) |
| oaa | observation azimuth angle | deg | 100 (195) |
| day | day | - | 14 |
| month | month | - | 7 |
| *Atmospheric conditions* | | | |
| - | atmospheric profile | - | US62 |
| *Aerosol type* | | | |
| - | aerosol model | - | Continental |
| *Altitudes* | | | |
| - | sensor altitude | km | 1000 (on board of satellite) |
| - | target altitude | km | 0.25 |
| *Ground reflectance* | | | |
| - | homogeneous Lambertian | - | Green vegetation |

*Appendix C.2. SCOPE Fixed Parameters*

**Table A3.** Input parameters of the SCOPE model that were fixed during the global sensitivity analysis. Angles in brackets are related to the oblique view. Varied SCOPE parameters are listed in Table 2.

| Parameter | Definition | Unit | Value |
|---|---|---|---|
| *Leaf thermal* | | | |
| rho_thermal | broadband leaf reflectance in thermal range | - | 0.01 |
| tau_thermal | broadband leaf transmittance in thermal range | - | 0.01 |
| *Leaf biochemical* | | | |
| BallBerry0 | | - | 0.01 |
| Type | photochemical pathway (C3 or C4) | - | C3 |
| *Fluorescence* | | | |
| Tparam | temperature response of fluorescence | K | [0.2, 0.3, 281, 308, 328] |
| fqe | fluorescence quantum yield efficiency at photosystem level | - | 0.01 |
| *Soil* | | | |
| rs_thermal | broadband soil reflectance in the thermal range | - | 0.06 |
| cs | specific heat capacity of the soil | $J\,kg^{-1}\,K^{-1}$ | 1180 |
| rhos | specific mass of the soil | $kg\,m^{-3}$ | 1800 |
| *Meteo* | | | |
| Oa | atmospheric $O_2$ concentration | per mille | 209 |
| *Aerodynamic* | | | |
| CR | drag coefficient for isolated tree | - | 0.35 |
| CD1 | fitting parameter | - | 20.6 |
| Psicor | roughness layer correction | - | 0.2 |
| CSSOIL | drag coefficient for soil | - | 0.01 |
| rb | leaf boundary resistance | $s\,m^{-1}$ | 10 |
| *Angles* | | | |
| tts | solar zenith angle | deg | 50 |
| tto | observations zenith angle | deg | 22 (50) |
| psi | relative azimuth angle | deg | 130 (135) |

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
