# Peer review of "Global Sensitivity Analysis of the SCOPE Model in Sentinel-3 Bands: Thermal Domain Focus"

_remotesensing, doi:10.3390/rs11202424_

Round 1

Reviewer 1 Report

Dear authors,

Let me first acknowledge the work that you submitted to Remote Sensing. I find the topic highly interesting for the readers, with a very complete analysis of the sensitivity of OLCI and SLSTR bands in the entire optical range to atmospheric and vegetation biophysical variables.

Once said this, I would recommend that you revise the structure of your paper, particularly with respect to the Annexes presented here. In my opinion, some of the Annexes could be merged within the text, which would facilitate readers to go through the document. Also, some of the figures presented in the Annex lack from supportive explanatory text that, together with the small size of the figure and amount of information contained here, makes it difficult to understand their content.

I would appreciate if you could go through the following minor revisions/comments.

Thanks again for this interesting work.

Section 1: as a general comment, I find that this section has many small paragraphs and this reduces the readibility and logical structure of the introduction. Please consider to rearrange/merge paragraphs containing a similar message. E.g.1: paragraphs at L33, L39, L47 and L53 could be considered as examples of GSA applied on radiative trasnfer models. E.g.2: paragraphs at L60, L66, L70 and L74 are talking about previous studies (and limitation) of GSA on SCOPE model

L36: use citation "Sun et al. [6]" instead of "by [6]". Please review other similar citations

L82-88: please add reference (with hyperlink) to sections of the paper organization. E.g.: "materials and methods (Section 2)...". This will facilitate to navigate through the document

L104: I guess RTMo and RTMt correspond to the optical and thermal domain. If that's the case, please consider adding these RTMo and RTMt in parenthesis after "optical" and "thermal" for clarification of the following subsections

L108: out of curiosity, could SCOPE have different values of the leave optical properties within the canopy? (e.g. varying Chlorophyll concentration between a reasonable min-max range with a given statistical and spatial distribution)

L117: please consider to merge this paragraph with the previous one. In fact, please review the entire manuscript taking into account this comment

Figure 1: legends, axis and titles are very small. Please increase font size and use the entire paper width to increase the size of the figure.

Appendix B1: please change the current title "6S" to something more descriptive of the content of this annex

Equation 3, L178-L180 and Annex B1: There is something that I don't understand in these equations (including annex). Let me explain: from the Lambertian approximation, one can express the TOA radiance as

Ltoa = Lp + (1/π)*Es*Tg*Tdown*Tup*ρ/(1-S*ρ)

where Lp is the path radiance, Tdown and Tup are the total transmittance in the Sun-to-Target and Target-to-Sensor paths, Tg is the gas tranmittance, S is the spherical albedo and Es is the solar irradiance
The atmospheric path radiance (Lp) can be expressed as:
Lp = ρp*Es*Tg
being ρp the atmospheric path reflectance. which would be equivalent to your factor xb/xa. Accordingly, xb is not the path radiance as you defined in L179.
As a general comment, I think that the way you described your TOA radiance simulations is overly complex and it leads to misunderstanding. You could simply rephrase these lines and equation 3 (also removing the entire Annex B1), simply by saying that the TOA radiance simulation is made following the Lambertian equation:
Ltoa = Lp + (1/π)*Es*Tg*Tdown*Tup*ρ/(1-S*ρ)
where
Lp = ρp*Es*Tg
where all these terms ( ρp, Es, Tg, Tdown, Tup and S) are provided by 6S as a direct output (in the .out files).

L190: if I'm not mistaken, 6S does not provide as output the BOC irradiance splitted into its direct and diffuse components. From the equations A1 and A2 in the Annex B1, I follow the logic to calculate these two components, however they come at expenses of knowing the diffuse transmittance (td) and the aot, which again are not direct outputs of 6S. Could you please explain how you obtained td and aot? Maybe Annex B1 could focus on this aspect

Equation 6 and 7: these equations could be written within the paragraph below (L211-L212) to economise space and give more important to equation 5, which is the one more relevant here

L249: I would recommend to put the figure A5 in the same page where is referenced rather than on the Annex A5

Figure 3: It would be advisable to use the full width of the paper to have a larger figure of the GSA

L270: before moving to the analysis on OLCI and SLSTR, could you please comment these results and compare them with those is https://doi.org/10.3390/rs11161923 ? Results seem to be compatible but some differences appear, particularly on the higher influence of AOT in your results (maybe due to a higher variation (0-1) than in Verrelst paper (0-0.4)?) and the lower influence of water vapor (maybe since 6S does not couple absorption and scattering?). Consider adding this suggestion as comments in Section 4

Figure 4: same comment as for Figure 3 (and others in the manuscript), use the full width of the page to make the figure wider and more readible

Figures A1 to A3: in my opinion, these figures can be removed from the manuscript. The low values of the non significant variables in the GSA make difficult to observe them and also there is no suportive text to explain these figures. In general, consider reducing the number of Annexes. In my opinion, instead of clarifying information, they make it more complex for the reader to follow all the information

Author Response

Dear reviewer,

Thank you for your time and valuable comments, which has made our work much-much better and helped me personally understand the atmospheric correction and 6S model.

If you would like to track changes, we suggest using Notepad++ Compare plugin and attach previous version of the .tex file (Sensitivity_paper_for_track_change.tex) for comparison.

Please, find point-by-pint reply to your comments below.

Best regards,

Egor Prikaziuk

Section 1: as a general comment, I find that this section has many small paragraphs and this reduces the readibility and logical structure of the introduction. Please consider to rearrange/merge paragraphs containing a similar message. E.g.1: paragraphs at L33, L39, L47 and L53 could be considered as examples of GSA applied on radiative trasnfer models. E.g.2: paragraphs at L60, L66, L70 and L74 are talking about previous studies (and limitation) of GSA on SCOPE model

            Done. These paragraphs were indeed short. They were merged as suggested.

L36: use citation "Sun et al. [6]" instead of "by [6]". Please review other similar citations

            Done for this and similar citations. Current line number is L34.

L82-88: please add reference (with hyperlink) to sections of the paper organization. E.g.: "materials and methods (Section 2)...". This will facilitate to navigate through the document

            Done, hyperlinks for sections were added in paragraph L80-86

L104: I guess RTMo and RTMt correspond to the optical and thermal domain. If that's the case, please consider adding these RTMo and RTMt in parenthesis after "optical" and "thermal" for clarification of the following subsections

The abbreviations are now spelled out now in Line 101 (RTMo) and Line 102 (RTMt) and as headers of sections 2.1.1 (Line 104) and 2.1.2 (Line 116). Please, note the added comments in lines 101-103 that RTMo works with incident radiation (both optical and thermal) and RTMt with emitted thermal radiation.

L108: out of curiosity, could SCOPE have different values of the leave optical properties within the canopy? (e.g. varying Chlorophyll concentration between a reasonable min-max range with a given statistical and spatial distribution)

This is not a standard option in SCOPE although it could be coded. The alternative model mSCOPE (Yang et al, 2017) allows for this type of within-canopy heterogeneity, but SCOPE does not.

L117: please consider to merge this paragraph with the previous one. In fact, please review the entire manuscript taking into account this comment

done, merged many paragraphs in methods section. Discussion is left unchanged as each paragraph discusses separate aspect / model variable.

Figure 1: legends, axis and titles are very small. Please increase font size and use the entire paper width to increase the size of the figure.

Improved

Appendix B1: please change the current title "6S" to something more descriptive of the content of this annex

The current title is: ‘Appendix B.1. Atmospheric correction in 6S atmospheric model’ (L515). In fact this appendix was seriously revised and shortened, as you recommended in the following comments. Table with 6S output was added into subsection ‘Appendix B.2. Output parameters of 6S atmospheric model’ (L519).

Equation 3, L178-L180 and Annex B1: There is something that I don't understand in these equations (including annex). Let me explain: from the Lambertian approximation, one can express the TOA radiance as

Ltoa = Lp + (1/π)*Es*Tg*Tdown*Tup*ρ/(1-S*ρ)

where Lp is the path radiance, Tdown and Tup are the total transmittance in the Sun-to-Target and Target-to-Sensor paths, Tg is the gas tranmittance, S is the spherical albedo and Es is the solar irradiance
The atmospheric path radiance (Lp) can be expressed as:
Lp = ρp*Es*Tg
being ρp the atmospheric path reflectance. which would be equivalent to your factor xb/xa. Accordingly, xb is not the path radiance as you defined in L179.

Indeed, there was a mistake in xb coefficient formula: I missed Tg term in the denominator. However, the equations for xa and xc were correct. Now they are labelled as Equation A2 (xa), A3 (xb), A4 (xc). The shortest explanation of xb coefficient is xb = Latm * xa.

xa = π / (Es* µs * Tg*Tdown*Tup)

As a general comment, I think that the way you described your TOA radiance simulations is overly complex and it leads to misunderstanding. You could simply rephrase these lines and equation 3 (also removing the entire Annex B1), simply by saying that the TOA radiance simulation is made following the Lambertian equation:
Ltoa = Lp + (1/π)*Es*Tg*Tdown*Tup*ρ/(1-S*ρ)
where 
Lp = ρp*Es*Tg
where all these terms ( ρp, Es, Tg, Tdown, Tup and S) are provided by 6S as a direct output (in the .out files).

Done, with except that Tdown * Tup are taken together as Ts (total scattering transmittance) and µs - cos(SZA) - added. Now Equation(3) is in accordance with your suggestion. 6S output names for Latm, Es, Tg, Ts, S are given in TableA1. The same equation in terms of 6S coefficients xa, xb, xc was moved to appendices: Equation A1.

L190: if I'm not mistaken, 6S does not provide as output the BOC irradiance splitted into its direct and diffuse components. From the equations A1 and A2 in the Annex B1, I follow the logic to calculate these two components, however they come at expenses of knowing the diffuse transmittance (td) and the aot, which again are not direct outputs of 6S. Could you please explain how you obtained td and aot? Maybe Annex B1 could focus on this aspect

In fact, 6S does provide direct and diffuse BOA irradiance, but does not provide aot and diffuse transmittance. We wanted to show how BOA irradiance is calculated and provided the equations. However, we fully agree with your last comment that in the way it was the appendix confused readers rather than clarified the approach, so we decided to remove these formulas and substitute them with a table where 6S output name is linked to the symbol we use throughout the paper.

Equation 6 and 7: these equations could be written within the paragraph below (L211-L212) to economise space and give more important to equation 5, which is the one more relevant here

Done, now they are in lines 201

L249: I would recommend to put the figure A5 in the same page where is referenced rather than on the Annex A5

Done, now it is Figure 3 on page 10

Figure 3: It would be advisable to use the full width of the paper to have a larger figure of the GSA

Done for all images [width=1\linewidth]

L270: before moving to the analysis on OLCI and SLSTR, could you please comment these results and compare them with those is https://doi.org/10.3390/rs11161923 ? Results seem to be compatible but some differences appear, particularly on the higher influence of AOT in your results (maybe due to a higher variation (0-1) than in Verrelst paper (0-0.4)?) and the lower influence of water vapor (maybe since 6S does not couple absorption and scattering?). Consider adding this suggestion as comments in Section 4

Done. Added to introduction L57-58. And paragraph L365-372. Also in Table 1 provided the range of ozone in atm-cm and water in g cm-2 as a standard unit for the ease of comparison.

Figure 4: same comment as for Figure 3 (and others in the manuscript), use the full width of the page to make the figure wider and more readable

Done for all images [width=1\linewidth]

Figures A1 to A3: in my opinion, these figures can be removed from the manuscript. The low values of the non significant variables in the GSA make difficult to observe them and also there is no suportive text to explain these figures. In general, consider reducing the number of Annexes. In my opinion, instead of clarifying information, they make it more complex for the reader to follow all the information

Done, appendices are rearranged in sections: results, methods and fixed parameters. Proper headers are given to each subsection.

Reviewer 2 Report

This study explores the possibilities of using the SCOPE model together with Sentinel-3 to exploit the OLCI and SLSTR in synergy. Sobol’ variance-based GSA of TOA radiance produced with a coupled SCOPE-6S model was conducted for optical bands of OLCI and SLSTR, while another GSA of SCOPE was conducted for the LST product of SLSTR. The results could provide instruction for the application of Sentinel-3 data on the extraction of vegetation parameters. Some details should be modified to make the paper more readable.

There are lots of abbreviation in the text. But some full spellings are missed. For example, what are the full spellings of RTMo in 2.1.1 and RTMt in 2.1.2? The description of some terms in Line 115-116 is not accurate. (bidirectional (BDRF), directional-hemispherical (DHRF), hemispherical-directional (HDRF), 116 bihemispherical (BHRF)) There are two 'was selected' in Line 157. The relationship between LTOA and ρTOC should be explained in Equation(3) Both of the two figures of Figure8. are labelled (b)

Author Response

Dear reviewer,

Thank you for your comments, point-by-point reply is presented below.

In addition to your recommendations we rearranged Appendices section and added one more recent work to discussion and introduction.

If you would like to track changes, we suggest using Notepad++ Compare plugin and attach previous version of the .tex file (Sensitivity_paper_for_track_change.tex) for comparison.

Best regards,

Egor Prikaziuk

There are lots of abbreviation in the text. But some full spellings are missed. For example, what are the full spellings of RTMo in 2.1.1 and RTMt in 2.1.2?

The abbreviations are now spelled out now in Line 101 (RTMo) and Line 102 (RTMt) and as headers of sections 2.1.1 (Line 104) and 2.1.2 (Line 116). Please, notice the comments in lines 101-103 that RTMo works with incident radiation (both optical and thermal) and RTMt with emitted thermal radiation.

The description of some terms in Line 115-116 is not accurate. (bidirectional (BDRF), directional-hemispherical (DHRF), hemispherical-directional (HDRF), 116 bihemispherical (BHRF))

Changed to Verhoef’s style (rso, rdo), link to Schaepman-Strub2006 remained for the clarity of table 2 in that paper. Now the line numbers are 113-114. Equation (1) was modified accordingly.

There are two 'was selected' in Line 157.

removed, now the line is 151.

The relationship between LTOA and ρTOC should be explained in Equation(3)

Improved, now Equation(3) includes commonly used terms: Latm, Es, Tg, Ts, S available from 6S output, with output names given in TableA1. Previously used equation in terms of 6S coefficients xa, xb, xc was moved to appendices: Equation A1.

Both of the two figures of Figure8. are labelled (b) 

corrected